# Latent Space Translation via Semantic Alignment

**Valentino Maiorca**[1,*]   **Luca Moschella**[1,*]

**Antonio Norelli**[1]   **Marco Fumero**[1]   **Francesco Locatello**[2]   **Emanuele Rodolà**[1]

[1]Sapienza University of Rome   [2]Institute of Science and Technology Austria (ISTA)

## Abstract

While different neural models often exhibit latent spaces that are alike when exposed to semantically related data, this intrinsic similarity is not always immediately discernible. Towards a better understanding of this phenomenon, our work shows how representations learned from these neural modules can be translated between different pre-trained networks via simpler transformations than previously thought. An advantage of this approach is the ability to estimate these transformations using standard, well-understood algebraic procedures that have closed-form solutions. Our method directly estimates a transformation between two given latent spaces, thereby enabling effective stitching of encoders and decoders without additional training. We extensively validate the adaptability of this translation procedure in different experimental settings: across various trainings, domains, architectures (e.g., ResNet, CNN, ViT), and in multiple downstream tasks (classification, reconstruction). Notably, we show how it is possible to zero-shot stitch text encoders and vision decoders, or vice-versa, yielding surprisingly good classification performance in this multimodal setting.

## 1   Introduction

Representation learning [Bengio et al., 2014] is a fundamental paradigm in the field of artificial intelligence, aimed at uncovering the underlying structure of complex data. One of the main goals of representation learning is to discover a robust representation of the data that is insensitive to certain transformations of the input. The Manifold Hypothesis [Fefferman et al., 2013] posits that real-world data lies on a low-dimensional non-linear manifold embedded in a high-dimensional space. Yet, a complication arises in modeling these non-linear manifolds: the learning process is usually influenced by stochasticities in the training dynamics and extrinsic factors that do not pertain to the data's core attributes, resulting in different representations for samples expected to be similar (e.g., different views of the same object, multiple translations of the same sentence, or even the exact same input sample). This is critical as it hinders knowledge transfer between these networks. Recently, the concept of relative representations [Moschella et al., 2023] has been proposed as a method for zero-shot communication between latent spaces that is invariant to these extrinsic factors. The idea is that latent spaces of neural networks trained on comparable data can be projected into the same relative space, derived from the distances between the data points. One of the main contributions of relative encoding is that it shows how the signal encoded in the *angles* with respect to a reduced set of data points (called *anchors*) is enough to capture the intrinsic shape of the latent space, reaching results on various benchmarks comparable to those using the original (absolute) encodings. As a consequence, they empirically demonstrate that different latent spaces that share the same data semantics (i.e., different representations of the same high-level concepts, such as images and their captions), mostly differ only by an angle-preserving transformation.

---

[*]Equal contribution.

37th Conference on Neural Information Processing Systems (NeurIPS 2023).

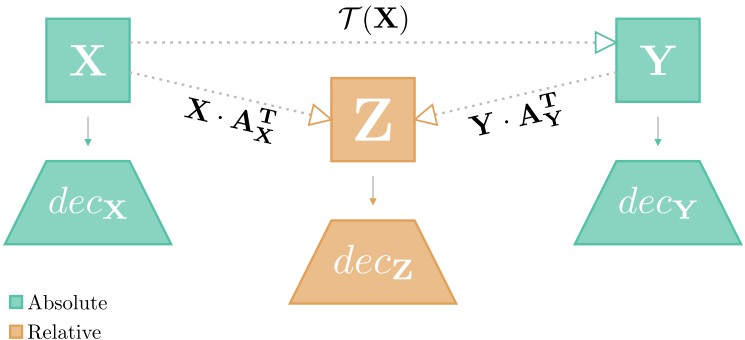

Figure 1: Zero-shot stitching of $\mathbf{X}$ and $\mathbf{Y}$ absolute spaces utilizing relative representations and our method (the estimation of $\mathcal{T}$). Our approach does not require a decoder specifically trained on relative representations ($dec_{\mathbb{Z}}$). Instead, we directly translate latent spaces, enabling the use of arbitrarily pre-trained decoders originally trained on absolute spaces.

Building on this intuition of the existence of a relatively simple transformation, we show the effectiveness and applications of *directly translating between different latent spaces* – provided that a partial (and possibly sparse) correspondence between data points is given. Remarkably, the process of seamlessly combining different neural networks, pre-trained on diverse datasets, modalities, architectures, domains and downstream tasks, proves unexpectedly straightforward. For instance, we show how it enables the ability to effectively integrate any pre-trained text encoder with any image classification head, and vice versa, without requiring any additional re-training or assumptions (indeed, Moschella et al. [2023] assumes the decoders are trained on relative representations). The method difference is emphasized in Figure 1. While zero-shot stitching with relative representations assumes the use of a single decoder specifically trained on a relative space, our method permits the reuse of the decoders originally trained on the absolute spaces.

Our main contributions can be summarized as follows:

- We explore the direct translation between latent spaces of distinct neural networks to enable *latent communication*. In particular, leveraging a semantic correspondence in the data, we directly translate for the first time across different trainings, architectures, and modalities. Notably, we obtain excellent stitching performances even in cross-modal settings, where we apply arbitrary text classifiers on top of pre-trained image encodings (and vice-versa).

- We show that different downstream tasks, namely classification and generation, require modeling different transformations to obtain the most out of the translation between their latent spaces;

## 2    Related Works

**Representations similarity**    Recent years have witnessed a growing consensus among researchers in the deep learning community that effective neural networks tend to learn similar representations for semantically similar data, regardless of the architecture, task, or domain in which they are applied. This idea is supported by a plethora of empirical studies [Moschella et al., 2023, Norelli et al., 2023, Morcos et al., 2018, Li et al., 2016, Kornblith et al., 2019, Bonheme and Grzes, 2022, Tsitsulin et al., 2020, Barannikov et al., 2022, Vulić et al., 2020a, Lample et al., 2018, Lenc and Vedaldi, 2015, Mikolov et al., 2013a, Antonello et al., 2021, Bengio et al., 2012, Movshovitz-Attias et al., 2017, Chang et al., 2022] and the phenomenon is particularly pronounced for large and wide models [Somepalli et al., 2022, Mehta et al., 2022]. Nevertheless, despite this intrinsic similarity, latent spaces can still exhibit extrinsic variations. Our work analyzes the possibility of translating these spaces from one to another, linking these extrinsic variations to different classes of transformation.

**Manifold alignment**    Procrustes analysis has been instrumental in the alignment of latent spaces in deep neural networks [Wang and Mahadevan, 2008, 2009], particularly in Natural Language Processing (NLP) where it is well-known that latent spaces of different languages are isomorphic [Vulić et al., 2020b] and can be effectively aligned [Mikolov et al., 2013b, Xing et al., 2015]. Rooted

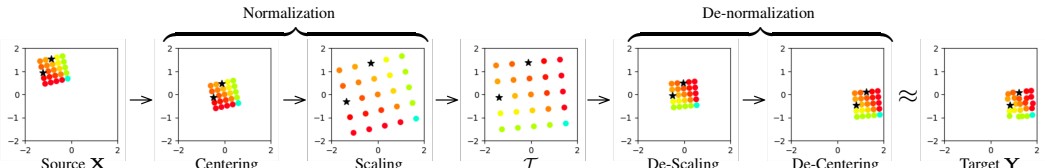

Figure 2: Method illustration on a synthetic example. Given a source space $\mathbf{X}$, the steps to translate it to a target $\mathbf{Y}$ are sequentially applied as described in Section 3.2. Note that the translation is not perfect due to an arbitrary distortion of the data.

in shape analysis, this method efficiently uncovers correspondences between latent spaces of different models through the estimation of an optimal orthogonal transformation [Gower, 1975]. Previous works largely exploit Procrustes analysis to align latent spaces originating from models of the same architecture [Csiszarik et al., 2021], such as multi-lingual FastText embeddings [Bojanowski et al., 2017, Smith et al., 2017]. Instead, in this work, we extend the application of Procrustes analysis to network stitching in new domains, architectures, and even modalities for multiple downstream tasks.

**Stitching and zero-shot**   Model stitching, which involves the combination of different neural networks to create a new model, has been a topic of active research in the field of representation learning. A key concept in this area is that of relative representations [Moschella et al., 2023, Norelli et al., 2023], which enables zero-shot stitching between different neural networks trained on semantically similar data. While this approach assumes the use of decoders *trained on relative representations*, our work removes this constraint by introducing a zero-shot mechanism for translating one absolute space to another without relying on a shared (relative) representation, enabling the stitching of arbitrarily trained models, further generalizable by assuming only the positive scale invariance of their decoder part. Previously, trainable stitching layers [Lenc and Vedaldi, 2015, Bansal et al., 2021, Csiszarik et al., 2021] have been introduced to allow for the combination of parts of different networks or to verify statements regarding latent space similarity. Other works [Gygli et al., 2021, an, 2021, Yaman et al., 2022, an, 2020] have proposed alternative methods for producing directly compatible and reusable network components without specific stitching layers. Here, we sidestep the need to create a new compatible representation and instead focus on obtaining a direct transformation to map from a source space to a target one to enable seamless network stitching. Concurrently to this work, a similar approach by [Lähner and Moeller, 2023] also targeted the direct alignment of representational spaces, focusing on the compatibility of models trained end-to-end.

## 3 Method

### 3.1 Preliminaries

Relative representation is a framework introduced in Moschella et al. [2023], which enables latent spaces of arbitrary neural models to communicate with each other. This is obtained by projecting the latent spaces into a common one, transitioning from an absolute coordinate frame to a **relative space**: each sample is represented as a function of a set of fixed samples denoted as **anchor set**. Specifically, the new representation is computed by independently projecting each sample point $\mathbf{x}$ in the latent space $\mathbf{X} \in \mathbb{R}^{n \times d}$, into the anchor set $\mathbf{A_X} \subset \mathbf{X}$. Formally, this is represented as

$$\mathbf{X}_{rel} = \mathbf{X}_{abs} \cdot \mathbf{A_X}^T, \tag{1}$$

where $\mathbf{X}_{rel} \in \mathbb{R}^{n \times k}, \mathbf{X}_{abs} \in \mathbb{R}^{n \times d}$ and $\mathbf{A_X} \in \mathbb{R}^{k \times d}$ Samples in $\mathbf{X}$ and in $\mathbf{A}$ are rescaled to unit norm, i.e. $\mathbf{x} = \frac{\mathbf{x}}{\|\mathbf{x}\|_2} \ \forall \mathbf{x} \in \mathbf{X}$ and $\mathbf{a} = \frac{\mathbf{a}}{\|\mathbf{a}\|_2} \ \forall \mathbf{a} \in \mathbf{A_X}$.

We assume to have access to subsets $\mathcal{A_X} \subset \mathcal{X}$ and $\mathcal{A_Y} \subset \mathcal{Y}$, with $\mathcal{X}$ and $\mathcal{Y}$ being the data distributions, and that there exists a correspondence $\Gamma : \mathcal{A_X} \mapsto \mathcal{A_Y}$ between these two sets of *parallel anchors*. Parallel anchors act as a "Rosetta stone" [Norelli et al., 2023], meaning they establish a semantic correspondence between their respective spaces: an anchor sample in the first set represents the same high-level concept as its counterpart in the second set. This allows stitching together components of different models: i.e., merging independently trained encoder and decoder modules from different networks. The relative projection will map latent spaces into the same one as long as the core assumption that they differ by an angle-preserving transformation is satisfied. However, in

order to perform the stitching procedure in Moschella et al. [2023], decoders must be trained from scratch at least once to process samples in this shared relative space.

In this work, we overcome this need by substituting the costly retraining procedure with an efficient strategy to directly estimate the transformation necessary to map between spaces. Moreover, we relax the "angle-preserving" constraint by allowing for a broader class of transformations obtained via robust, closed-form algorithms.

## 3.2 Latent Space Translation

Consider two latent spaces, $\mathbf{X} \in \mathbb{R}^{n \times d_1}$ and $\mathbf{Y} \in \mathbb{R}^{n \times d_2}$. Our objective is to estimate the transformation $\mathcal{T}$ that translates $\mathbf{X}$ into $\mathbf{Y}$: $\mathbf{Y} = \mathcal{T}(\mathbf{X})$, exploiting the **semantic alignment** between the two spaces. Throughout this work, we identify two main steps in the translation process: pre-processing the spaces and estimating the transformation $\mathcal{T}$, as outlined in Figure 2.

**Pre-processing**    Generally, the two spaces may have different dimensionalities – in those cases, we zero-pad the smaller one to match the dimension of the other without changing its underlying structure [Williams et al., 2021]. Moreover, we standardize each feature in the encoding to have zero mean and unit variance (standard scaling) if not otherwise specified, whose statistics are computed only on the anchor sets for both source and target space, to perform the necessary de-normalization.

**Estimating $\mathcal{T}$**    In Moschella et al. [2023], it is empirically shown that the spaces mostly differ by an angle-preserving transformation. Nevertheless, we broaden our investigation by considering different ways of obtaining $\mathcal{T}$ to evaluate the robustness of that assumption and the versatility of our approach. Throughout our experiments, we primarily operate under the assumption that $\mathcal{T}$ can be constrained to encode, at most, an affine transformation: $\mathcal{T}(\mathbf{x}) = \mathbf{RX} + \mathbf{b}$

This general formulation, without additional constraints, corresponds to our `affine` method in the experiments, and it is optimized via gradient descent. The other transformations are trivially obtained by progressively adding constraints on this one:

- `linear`. To model a linear transformation, we can just set the bias term to zero $\mathbf{b} = \vec{0}$ and optimize via Least Square. Here we are both simplifying the class of transformations and switching from a gradient descent optimization to a closed-form procedure.
- `l-ortho`. Additionally, we could require $R$ to be orthogonal to encode an isometry. In this case, we obtain this by applying Singular Value Decomposition (SVD) on the corresponding $R$ obtained by the `linear` solution. Through this, we aim to understand the implications of enforcing orthogonality on a transformation that was originally not constrained to be so, in a setting similar to Xing et al. [2015].
- `ortho`. To obtain the optimal orthogonal $R$, we apply Procrustes analysis [Gower, 1975].

This methodology facilitates efficient and precise zero-shot translation between disparate latent spaces. The transformation $\mathcal{T}$, derived solely from the subset of corresponding points, provides a robust and versatile foundation for model reuse and interoperability in diverse machine learning contexts.

# 4    Latent Communication via Translation

In this section, we evaluate the capabilities and effectiveness of our translation method through various scenarios, highlighting its applicability in diverse contexts. We present empirical results in three different novel settings: i) cross-architecture; ii) cross-modality; iii) autoencoding. In each case, the translation performance of each method for obtaining the transformation $\mathcal{T}$ is evaluated against two baselines, the naive absolute one and the relative one.

**Stitching Procedure**    In line with the *zero-shot stitching* concept outlined in Moschella et al. [2023], we combine independent encoders and decoders (e.g., classifiers, generators) without subsequent training or fine-tuning. This study does not necessitate a decoder trained on relative representations; instead, we directly employ the original decoders trained on absolute spaces. Each benchmark we perform follows the same procedure unless otherwise specified: we measure the mean performance over all the possible combinations of (encoder, decoder) for each test set in different settings:

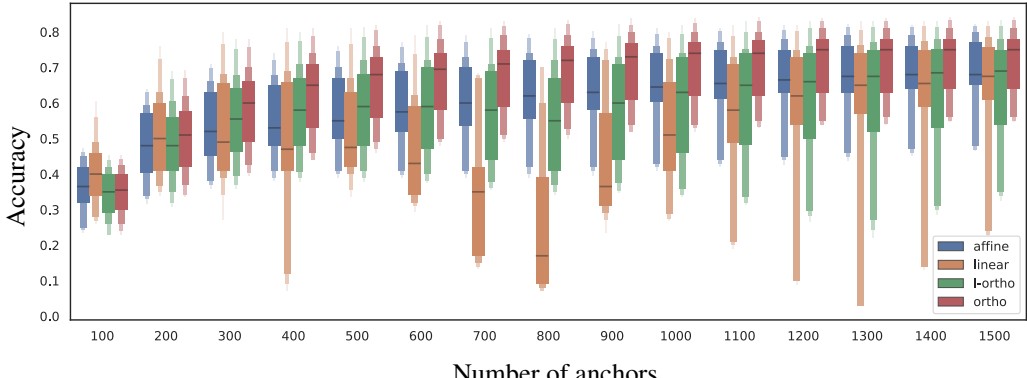

Figure 3: Performance comparison of `affine`, `linear`, `l-ortho`, and `ortho` at varying number of anchors on classification accuracy. Results on `CIFAR100` fine-grained. The same analysis for the generation case is in Figure 8 in the Appendix.

Table 1: Cross-architecture stitching with various methods for estimating $\mathcal{T}$ and applying standard scaling. The stitched decoders are SVMs with a linear kernel. 5 runs for each encoder-decoder pair. (C) and (F) next to `CIFAR100` indicate, respectively, coarse-grained and fine-grained. Please refer to the Appendix in Table 5 for additional results with MLPs as classification heads.

| | Dataset | No Stitching | absolute | relative | affine | linear | l-ortho | ortho |
|---|---|---|---|---|---|---|---|---|
| **Vision** | CIFAR10 | $0.95 \pm 0.03$ | $0.16 \pm 0.22$ | $0.80 \pm 0.22$ | $0.92 \pm 0.05$ | $0.88 \pm 0.11$ | $0.90 \pm 0.09$ | $0.93 \pm 0.04$ |
| | CIFAR100-C | $0.85 \pm 0.07$ | $0.11 \pm 0.21$ | $0.54 \pm 0.25$ | $0.78 \pm 0.09$ | $0.73 \pm 0.16$ | $0.77 \pm 0.11$ | $0.81 \pm 0.07$ |
| | CIFAR100-F | $0.76 \pm 0.09$ | $0.07 \pm 0.21$ | $0.30 \pm 0.24$ | $0.68 \pm 0.11$ | $0.62 \pm 0.19$ | $0.64 \pm 0.16$ | $0.71 \pm 0.09$ |
| | F-MNIST | $0.88 \pm 0.01$ | $0.15 \pm 0.20$ | $0.63 \pm 0.23$ | $0.86 \pm 0.01$ | $0.83 \pm 0.06$ | $0.82 \pm 0.05$ | $0.85 \pm 0.02$ |
| | MNIST | $0.96 \pm 0.01$ | $0.15 \pm 0.21$ | $0.50 \pm 0.22$ | $0.94 \pm 0.01$ | $0.89 \pm 0.08$ | $0.81 \pm 0.11$ | $0.91 \pm 0.02$ |
| **Text** | TREC | $0.87 \pm 0.12$ | $0.20 \pm 0.06$ | $0.36 \pm 0.13$ | $0.82 \pm 0.12$ | $0.74 \pm 0.25$ | $0.57 \pm 0.25$ | $0.79 \pm 0.11$ |
| | AG News | $0.73 \pm 0.09$ | $0.25 \pm 0.02$ | $0.39 \pm 0.13$ | $0.65 \pm 0.08$ | $0.62 \pm 0.08$ | $0.61 \pm 0.10$ | $0.66 \pm 0.10$ |
| | DBpedia | $0.78 \pm 0.23$ | $0.07 \pm 0.01$ | $0.16 \pm 0.10$ | $0.66 \pm 0.24$ | $0.62 \pm 0.23$ | $0.57 \pm 0.23$ | $0.66 \pm 0.22$ |
| | IMDB | $0.61 \pm 0.04$ | $0.50 \pm 0.01$ | $0.51 \pm 0.02$ | $0.59 \pm 0.04$ | $0.57 \pm 0.04$ | $0.56 \pm 0.03$ | $0.59 \pm 0.04$ |

- *no-stitch*. The end-to-end performance of the decoder applied to the original space it was trained on. This is useful to establish an upper-bound in performances;

- *absolute*. The result of using the encodings without any transformation, we consider this as a probe for any pre-existing compatibility among encodings and, therefore, a lower-bound;

- *translation*. These are the results of the application of our latent translation method, with the estimation of $\mathcal{T}$ via `affine`, `linear`, `l-ortho` and `ortho`.

In each instance, we use the same parallel anchors, that are uniformly chosen, in a quantity comparable with the dimensionality of the absolute representation.

## 4.1 Cross-Architecture

Firstly, we test our method in a cross-architecture setting, zero-shot stitching together encodings coming from a variety of pre-trained networks and their associated absolute decoders (classifiers). This scenario provides an extensive testing ground for our method and demonstrates its robustness across different architectures. Please refer to Table 8 in the Appendix for further results on cross-architecture stitching in generation tasks.

**Experimental setting** We consider a variety of Computer Vision (`MNIST`, `Fashion MNIST`, `N24News`, `CIFAR10`, `CIFAR100`) and Natural Language Processing (`TREC` [Hovy et al., 2001, Li and Roth, 2002], `DBpedia` [Auer et al., 2007], `N24News` [Wang et al., 2022], `AG News` [Zhang et al., 2015], `IMDB` [Maas et al., 2011]) datasets. For the text domain we consider 7 different language models as encoders (uncased and cased BERT [Devlin et al., 2019], Electra [Clark et al., 2020], RoBERTa base [Liu et al., 2019], ALBERT [Lan et al., 2020], and the text encoder of [Radford et al., 2021]), and for the image domain 6 encoders (RexNet100 [Han et al., 2020], 4 variations of ViT [Dosovitskiy et al., 2020], and the image encoder of [Radford et al., 2021]), all pre-trained and frozen.

Table 2: Cross-architecture stitching with various methods for estimating $\mathcal{T}$ and applying L2 normalization. The stitched decoders are SVMs with linear kernel. 5 runs for each encoder-decoder pair. (C) and (F) next to `CIFAR100` indicate, respectively, coarse-grained and fine-grained. Please refer to Table 6 in the Appendix for additional results with MLPs as classification heads.

| | Dataset | No Stitching | absolute | relative | affine | linear | l-ortho | ortho |
|---|---|---|---|---|---|---|---|---|
| **Vision** | CIFAR10 | $0.95 \pm 0.03$ | $0.16 \pm 0.22$ | $0.80 \pm 0.22$ | $0.93 \pm 0.04$ | $0.78 \pm 0.27$ | $0.88 \pm 0.12$ | $0.91 \pm 0.09$ |
| | CIFAR100-C | $0.85 \pm 0.07$ | $0.11 \pm 0.21$ | $0.54 \pm 0.25$ | $0.79 \pm 0.07$ | $0.65 \pm 0.25$ | $0.73 \pm 0.17$ | $0.79 \pm 0.10$ |
| | CIFAR100-F | $0.76 \pm 0.09$ | $0.07 \pm 0.21$ | $0.30 \pm 0.24$ | $0.69 \pm 0.10$ | $0.52 \pm 0.25$ | $0.62 \pm 0.19$ | $0.68 \pm 0.13$ |
| | F-MNIST | $0.88 \pm 0.01$ | $0.15 \pm 0.20$ | $0.63 \pm 0.23$ | $0.86 \pm 0.01$ | $0.65 \pm 0.23$ | $0.83 \pm 0.06$ | $0.84 \pm 0.05$ |
| | MNIST | $0.96 \pm 0.01$ | $0.15 \pm 0.21$ | $0.50 \pm 0.22$ | $0.94 \pm 0.01$ | $0.61 \pm 0.23$ | $0.90 \pm 0.08$ | $0.90 \pm 0.04$ |
| **Text** | TREC | $0.87 \pm 0.12$ | $0.20 \pm 0.06$ | $0.36 \pm 0.13$ | $0.82 \pm 0.12$ | $0.44 \pm 0.20$ | $0.74 \pm 0.23$ | $0.77 \pm 0.12$ |
| | AG News | $0.73 \pm 0.09$ | $0.25 \pm 0.02$ | $0.39 \pm 0.13$ | $0.66 \pm 0.08$ | $0.56 \pm 0.10$ | $0.62 \pm 0.08$ | $0.64 \pm 0.10$ |
| | DBpedia | $0.78 \pm 0.23$ | $0.07 \pm 0.01$ | $0.16 \pm 0.10$ | $0.66 \pm 0.24$ | $0.44 \pm 0.20$ | $0.62 \pm 0.23$ | $0.60 \pm 0.22$ |
| | IMDB | $0.61 \pm 0.04$ | $0.50 \pm 0.01$ | $0.51 \pm 0.02$ | $0.59 \pm 0.04$ | $0.55 \pm 0.03$ | $0.58 \pm 0.04$ | $0.59 \pm 0.04$ |

The full encoder list can be found in Table 7 in the Appendix. For each dataset and for each encoder, we train an SVM classification head (decoder) on top of their specific encodings. We then proceed with the standard stitching procedure outlined in Section 4 and collect the results. Please see Table 8 in the Appendix for cross-architecture stitching in generation tasks, where we extend this analysis by verifying that our method works even across autoencoders of different bottleneck sizes.

**Result analysis** The stitching results are in Table 1. As expected, the *absolute* encodings obtain a score comparable to random guessing while also considering fewer encoder combinations out of the possible ones due to the dimensionality mismatch between some of them. Notably, these results show that the transformation relating to these pre-trained encoders is indeed mostly orthogonal: i) `ortho` and `affine`, the narrowest and the broadest transformation classes considered, are the better-performing translation methods. But while the former is obtained via a simple and efficient closed-form algorithm, the latter is SGD-optimized (Section 3.2). ii) the `l-ortho` version improves or has small drops in performances over the `linear` transformation it is obtained from, confirming that the least squares procedure converges to an $\mathbf{R}$ which is almost orthogonal. Note that these results demonstrate the feasibility of combining pre-trained models without the need for retraining or fine-tuning, with negligible drops in performances across the board and without any additional assumption on the decoders. Please refer to Tables 5 and 6 in the Appendix for results with different decoders. In the Appendix (Figure 7), we extend the cross-architecture transfer to decoders trained on different domains (styles) of the same `CIFAR10` dataset: the original one and a grayscale one.

**Sensibility to Anchor Quantity** The number of anchors is an essential parameter in our approach. In Figure 3, we evaluate how the quantity of these anchors impacts the residual error and the overall performance of our method for this experimental setting. This analysis offers insights into the optimal number of anchors necessary for efficient latent space translation.

**Role of Scaling** Our approach is designed to accommodate generic (re)scaling methods as pre-processing steps. We advocate for the use of standard scaling, as it shows reliable performance in our experiments, indicating that the scale of the data points is useful in estimating the latent transformation $\mathcal{T}$. However, for completeness, we also consider L2 normalization, which is the standard normalization in relative representations. This normalization method generalizes the class of transformations handled by our method and introduces an element of complete scale invariance. It's important to note that when this level of generalization is introduced, a scale-invariant decoder is required since the norm information is effectively removed. In the relative representation work, this is implicitly accomplished by training a decoder on relative representations. In our setting, since we do not train the decoder, in this setting we just assume its scale invariance (more details in Appendix A.1). This investigation exemplifies the flexibility of our approach, capable of adapting to different normalization and pre-processing strategies based on the specific requirements of the task at hand. The results presented in Table 2, when compared with Table 1, indicate a stronger reliance of the text modalities on the information encoded in the norm. This is aligned with existing literature in the NLP domain [Oyama et al., 2022], which suggests that the scale of the encodings contains information (e.g., it is correlated with the token frequency).

These results in diverse scenarios showcase the flexibility and adaptability of our method, especially its robustness in translating between latent spaces of different dimensionality and domains.

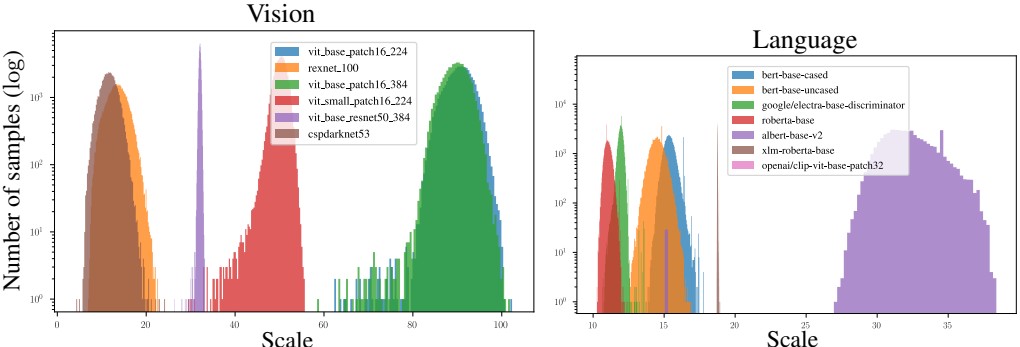

Figure 4: Scale distribution in encodings of different pre-trained encoders on the `N24News` dataset.

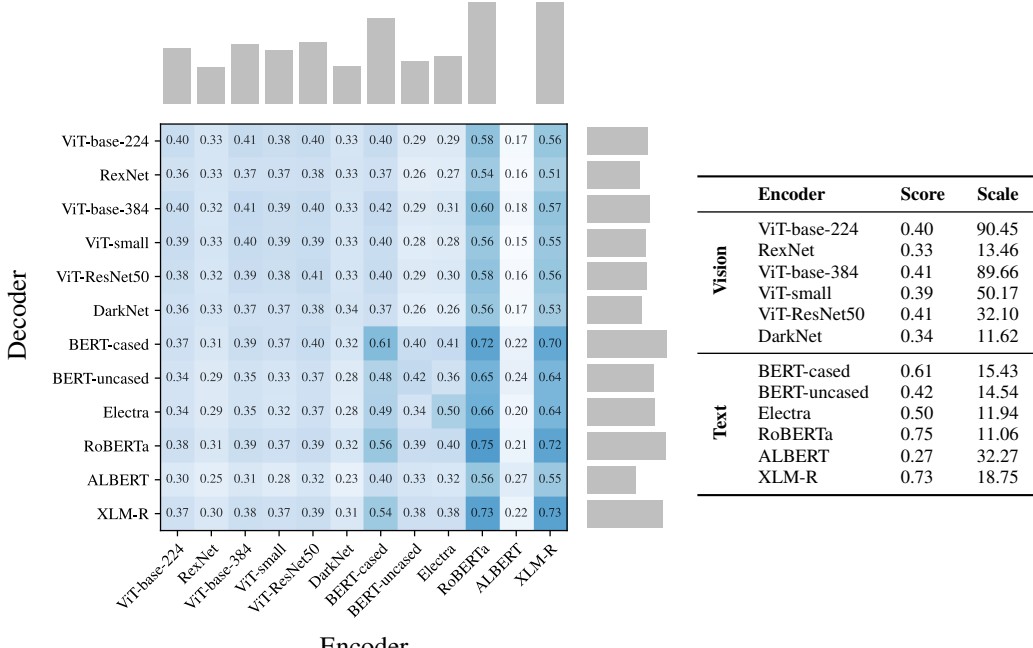

| | Encoder | Score | Scale |
|---|---|---|---|
| **Vision** | ViT-base-224 | 0.40 | 90.45 |
| | RexNet | 0.33 | 13.46 |
| | ViT-base-384 | 0.41 | 89.66 |
| | ViT-small | 0.39 | 50.17 |
| | ViT-ResNet50 | 0.41 | 32.10 |
| | DarkNet | 0.34 | 11.62 |
| **Text** | BERT-cased | 0.61 | 15.43 |
| | BERT-uncased | 0.42 | 14.54 |
| | Electra | 0.50 | 11.94 |
| | RoBERTa | 0.75 | 11.06 |
| | ALBERT | 0.27 | 32.27 |
| | XLM-R | 0.73 | 18.75 |

Figure 5: Performance comparison between different encoders and data modalities on the `N24News` multimodal dataset. On the right the accuracy of models trained end-to-end on a single data modality (Score) and their average norm (Scale). On the left the stitching performance between pairs of encoders and decoder. This shows the importance of translating from good encoders, that can even improve unimodal decoder performances. Results obtained with 2000 anchors and ortho, with an SVM as classification head. In the Appendix Figure 9, additional results using MLPs as decoders.

## 4.2 Cross-Modality

This scenario illustrates the applicability of our method in cross-modality settings, where we aim to translate between latent spaces of different modalities: text and image.

**Experimental setting** We adopt `N24News` [Wang et al., 2022], a multimodal news classification dataset that contains both text and associated pictures. We apply the standard encoding procedure to these two features separately, using different pre-trained uni-modal encoders. Then, we train a classification head (an SVM, please refer to Appendix Figure 9 for further results employing an MLP as classification head) on top of each one. Lastly, we zero-shot stitch each encoder with a classification head different from its corresponding one, measuring its classification accuracy, without further training or fine-tuning.

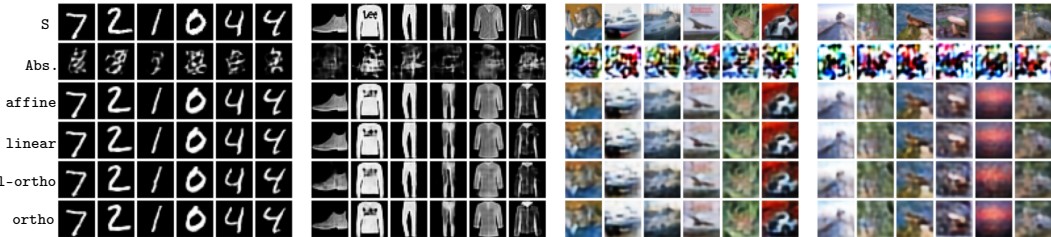

Figure 6: Reconstruction examples grouped by dataset. Each column is a different image, from top to bottom: original image, `absolute` stitching, `affine` stitching `linear` stitching, `l-ortho` stitching, and `ortho` stitching. No additional normalization applied on the decoder part. Please refer to Figures 10 and 11 in the Appendix for decoders trained with L2 normalization.

**Scale distributions**   In Figure 4, we present the scale distribution of the embeddings produced by several encoders on the `N24News` dataset. This empirical analysis shows a consistent pattern among encoders in that the scale distribution of their embeddings follows a Gaussian one with a single mode and a well-defined mean, which are usually compatible with standard scaling. This consistent behavior across encoders is likely attributed to their architectural choices, such as the normalization techniques, regularizations and the optimization problems they are designed to solve.

**Result analysis**   The discrepancy in the mean accuracy represented by the marginal bar plots in Figure 5 is a signal that can be used to identify spaces more suited to be *decoded into* and the ones that are stronger in *encoding from*. In fact, the language models as source space for the translation exhibit stronger performance than the vision encoders. We relate this behavior to the higher generality of the text domain data used during pre-training with respect to the image domain one [Zhai et al., 2022]. A remarkable finding in this setting is the improvement in classification performance when a modality-specific classifier trained on images is fed zero-shot with corresponding text encodings translated to the image domain via our method. This result underlines the significance of a good encoder and demonstrates the broad applicability of our technique. In practice, this means we can seamlessly apply image classifiers on textual data, and vice-versa.

These results show that our method: i) obtains effective zero-shot translation over different modalities; ii) improves unimodal decoders when translating from a better encoder than the one it was trained on.

### 4.3   Autoencoding

In this setting, our method is applied to align latent spaces of different trainings of the same autoencoder. The novelty of this scenario lies in the generation setting itself, as most prior works (Section 2) primarily focus on classification tasks. One key observation of [Cannistraci et al., 2023] is that the *task* at hand (e.g., classification, generation) defines a certain *class of transformations* (e.g. rotations) which act among the latent spaces. Restricting the search for the transformation to the right class, is fundamental in order to guarantee optimal performance and efficiency.

**Experimental setting**   We utilize four datasets for these experiments: `MNIST` [Lecun et al., 1998], `Fashion MNIST` [Xiao et al., 2017], and `CIFAR10` and `CIFAR100` Krizhevsky [2009]. For each dataset, we train two standard CNN-based autoencoder, with convolutions in the encoder and deconvolutions in the decoder, please refer to the supplementary material for further implementation details. The two autoencoders are identical in structure, differing only in the random seed used for weight initialization and data shuffling. To perform zero-shot stitching, we first translate each data point from the latent space of the first encoder to the latent space of the second using 1000 anchors. We then apply the second decoder to the translated data, without any further training or fine-tuning.

**Result analysis**   This experiment analyzes the alignment of latent spaces in different training regimens of the same autoencoder. The performance evaluation, as shown in Table 3, demonstrates that all methods `affine`, `linear`, `l-ortho`, and `ortho` yield satisfactory results. Moreover, qualitative results depicted in Figure 6 reveals minimal visual differences in the stitching outcomes across various datasets using different methods. Please refer to Figures 10 and 11 for other qualitative results. In fact, these results suggest that the latent spaces of image autoencoders are not exclusively correlated

Table 3: Zero-shot stitching for generation with various methods for estimating $\mathcal{T}$. Standard scaling used as normalization and the stitched decoders do not have any additional normalization. We report the latent cosine similarity (*lcos*) and MSE (*lmse*) between the target encoding and the translated one, but also the reconstruction MSE (*rmse*) between the input and the output. 1000 anchors used on 500 dimensional spaces. Please refer to Table 4 for results on decoders scale-invariant by design (with L2 normalization on the encodings).

|  | MNIST | | | Fashion MNIST | | | CIFAR10 | | | CIFAR100 | | |
|---|---|---|---|---|---|---|---|---|---|---|---|---|
|  | *lcos* | *lmse* | *rmse* | *lcos* | *lmse* | *rmse* | *lcos* | *lmse* | *rmse* | *lcos* | *lmse* | *rmse* |
| absolute | 0.09 | 0.27 | 0.14 | 0.17 | 0.23 | 0.23 | 0.30 | 0.29 | 0.34 | 0.34 | 0.53 | 0.40 |
| affine | 0.94 | 0.08 | 0.02 | 0.94 | 0.06 | 0.03 | 0.96 | 0.03 | 0.05 | 0.96 | 0.04 | 0.05 |
| linear | 0.92 | 0.09 | 0.02 | 0.93 | 0.07 | 0.04 | 0.94 | 0.03 | 0.05 | 0.94 | 0.04 | 0.06 |
| l-ortho | 0.79 | 0.14 | 0.02 | 0.78 | 0.12 | 0.05 | 0.85 | 0.05 | 0.06 | 0.84 | 0.07 | 0.07 |
| ortho | 0.90 | 0.10 | 0.02 | 0.90 | 0.08 | 0.04 | 0.94 | 0.03 | 0.06 | 0.93 | 0.04 | 0.06 |

by orthogonal transformations. Therefore, further research is warranted to explore and model the specific class of transformations that govern the correlation between neural networks during image autoencoding to constrain and enhance their approximation. For additional results pertaining to decoders with L2 normalization on their input, we refer to the Table 4 in the Appendix.

Overall these results, combined with Cannistraci et al. [2023] and Section 4.1, confirm that latent spaces in image autoencoders trained end-to-end are related by a class of transformations larger than orthogonal transformations.

# 5 Conclusion

At the heart of the proposed latent space translation lies the synergy between the principles of relative representation and classic algebraic techniques. The efficacy of this approach surpasses that of relative representations, emphasizing the benefits of directly estimating a transformation between specific latent space pairs instead of independently projecting them to a common one. This distinction underscores our contribution: we repurpose well-established techniques to serve as a translator across multiple latent spaces, enhancing efficiency in representation learning. With an extensive analysis of its applications in model reuse, we obtain a smooth compositionality of neural network modules across diverse computational frameworks, including those employing pre-trained models. Essentially, this paper showcases the adaptability and efficiency of manifold alignment methods in the emerging domain of zero-shot model compositionality.

**Future works and limitations** As with any new approach, there are limitations that warrant further exploration of our proposed method. For example, the optimal number of anchor points required for different tasks and datasets to boost performances, investigating the factors that could be linked to latent space compatibility (e.g., their intrinsic dimension), trade-offs between the granularity of the anchor set and its condition number. These are exciting research directions that we believe hold great potential for advancing the field and improving the effectiveness and robustness of our method.

**Acknowledgements** This work is supported by the ERC grant no.802554 (SPECGEO), PRIN 2020 project no.2020TA3K9N (LEGO.AI), and PNRR MUR project PE0000013-FAIR. Francesco Locatello did not contribute to this work at Amazon.

**Reproducibility Statement** We refer to the supplementary material for implementation details that are not described here in the main manuscript. Moreover, we release a modular PyTorch codebase[2] implementing the various translation methods and scaling techniques. All the experiments are carried out in deterministic environments to enable reproducibility, and the necessary data is versioned via DVC [Kuprieiev et al., 2022].

---

[2]https://github.com/Flegyas/latent-translation

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

# A   Additional results

In Figure 9, we present the outcomes of the multimodal experiment with an MLP employed as the classification head. The findings highlight the MLP's capability to leverage cross-modal information, leading to improved performance. However, the underlying mechanisms responsible for this enhancement remain unclear and warrant further investigation.

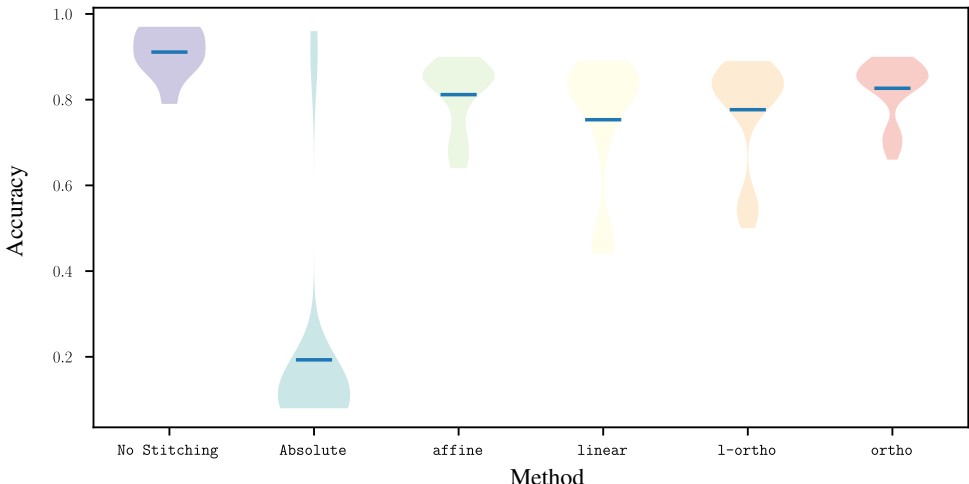

Figure 7: Cross-domain stitching on CIFAR10 and grayscale CIFAR10. 84 stitched pairs (pre-trained encoder - SVM classifier) for 5 different seeds.

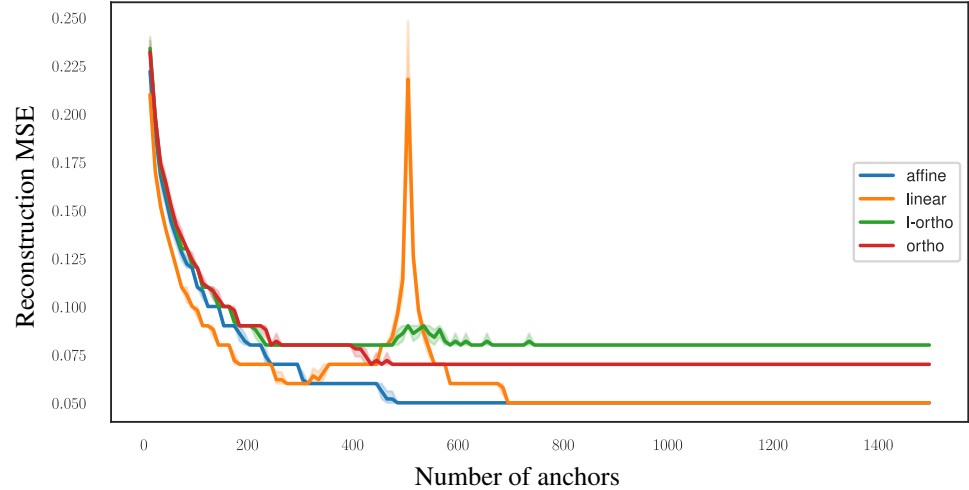

Figure 8: Performance comparison (reconstruction error) of `affine`, `linear`, `l-ortho` and `ortho` at varying anchor number on reconstruction task. Results on stitching 2 different `CIFAR100`-trained AEs with 5 samplings for each anchor quantity. The naive absolute baseline is flat on 0.38 as mean.

In Tables 5 and 6 quantitative results for stitching of MLP classifiers (differently from the main manuscript where SVMs are used) trained on top of pre-trained feature extractors, with and without additional L2 normalization, respectively.

In Figures 10 and 11, there are additional reconstruction examples with the same autoencoding setting as in the main manuscript, and with additional L2 normalization, respectively.

In Table 4 more quantitative results for stitching of autoencoders, with added L2 normalization (at training time) to the decoders of the reconstruction setting of the main manuscript.

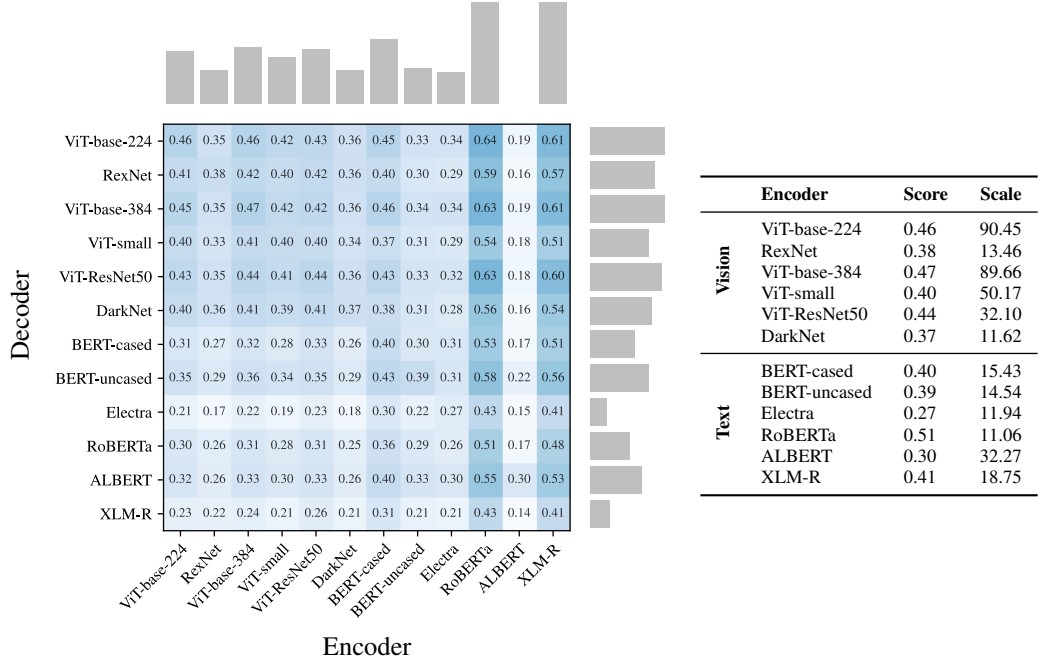

Figure 9: Performance comparison between different encoders and data modalities on the `N24News` multimodal dataset. On the right, the accuracy of models trained end-to-end on a single data modality (Score) and their average norm (Scale). On the left the stitching performance between pairs of encoders and decoder. This shows the importance of translating from good encoders, that can even improve unimodal decoder performances. Results obtained with 2000 anchors and `SVD`, with a MLP as classification head.

| | Encoder | Score | Scale |
|---|---|---|---|
| **Vision** | ViT-base-224 | 0.46 | 90.45 |
| | RexNet | 0.38 | 13.46 |
| | ViT-base-384 | 0.47 | 89.66 |
| | ViT-small | 0.40 | 50.17 |
| | ViT-ResNet50 | 0.44 | 32.10 |
| | DarkNet | 0.37 | 11.62 |
| **Text** | BERT-cased | 0.40 | 15.43 |
| | BERT-uncased | 0.39 | 14.54 |
| | Electra | 0.27 | 11.94 |
| | RoBERTa | 0.51 | 11.06 |
| | ALBERT | 0.30 | 32.27 |
| | XLM-R | 0.41 | 18.75 |

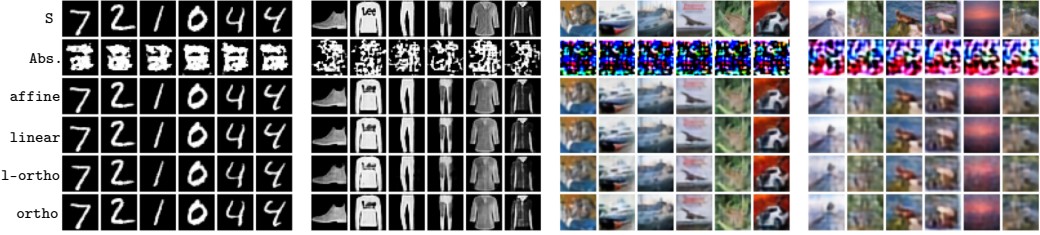

Figure 10: Reconstruction examples grouped by dataset. Each column is a different image, from top to bottom: original image, absolute stitching, `LSS` stitching, `OLSS` stitching, and `SVD` stitching. An L2 normalization is applied to the decoder input.

Table 4: Zero-shot stitching for generation. With `SVD` for estimating $\mathcal{T}$ and standard scaling as pre-processing. An L2 normalization is applied to the decoder input. We report the latent cosine similarity (*lcos*) and MSE (*lmse*) between the target encoding and the translated one, but also the reconstruction MSE (*rmse*) between the input and the output.

| | MNIST | | | Fashion MNIST | | | CIFAR10 | | | CIFAR100 | | |
|---|---|---|---|---|---|---|---|---|---|---|---|---|
| | *lcos* | *lmse* | *rmse* | *lcos* | *lmse* | *rmse* | *lcos* | *lmse* | *rmse* | *lcos* | *lmse* | *rmse* |
| `Abs.` | 0.39 | 0.98 | 0.28 | 0.53 | 0.97 | 0.33 | 0.62 | 1.23 | 0.46 | 0.59 | 1.17 | 0.38 |
| `affine` | 0.99 | 0.15 | 0.01 | 0.99 | 0.16 | 0.03 | 0.99 | 0.16 | 0.04 | 0.99 | 0.12 | 0.05 |
| `linear` | 0.98 | 0.17 | 0.01 | 0.98 | 0.18 | 0.03 | 0.99 | 0.16 | 0.04 | 0.99 | 0.13 | 0.05 |
| `l-ortho` | 0.89 | 0.41 | 0.02 | 0.91 | 0.41 | 0.04 | 0.96 | 0.39 | 0.05 | 0.93 | 0.30 | 0.08 |
| `ortho` | 0.97 | 0.21 | 0.02 | 0.97 | 0.23 | 0.03 | 0.99 | 0.21 | 0.05 | 0.96 | 0.22 | 0.07 |

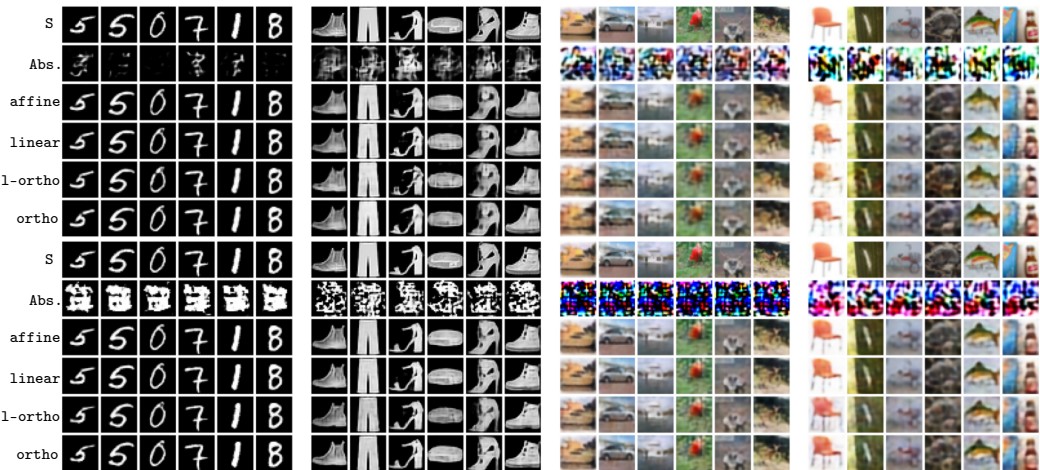

Figure 11: Additional reconstruction examples grouped by dataset. Each column is a different image, from top to bottom: original image, absolute stitching, `LSS` stitching, `OLSS` stitching, and `SVD` stitching. In the first row, no additional normalization is applied on the decoder input; in the second row, an L2 normalization is applied instead.

Table 5: Cross-architecture stitching with various methods for estimating $\mathcal{T}$ and employing standard scaling. The stitched decoders are simple MLPs. 5 runs for each encoder-decoder pair. (C) and (F) next to `CIFAR100` indicate, respectively, coarse-grained and fine-grained.

| | Dataset | No Stitching | absolute | relative | affine | linear | l-ortho | ortho |
|---|---|---|---|---|---|---|---|---|
| **Vision** | CIFAR10 | $0.95 \pm 0.03$ | $0.16 \pm 0.22$ | $0.73 \pm 0.21$ | $0.93 \pm 0.05$ | $0.89 \pm 0.11$ | $0.90 \pm 0.09$ | $0.93 \pm 0.04$ |
| | CIFAR100-C | $0.82 \pm 0.07$ | $0.11 \pm 0.21$ | $0.39 \pm 0.17$ | $0.76 \pm 0.08$ | $0.71 \pm 0.15$ | $0.74 \pm 0.11$ | $0.78 \pm 0.07$ |
| | CIFAR100-F | $0.68 \pm 0.14$ | $0.06 \pm 0.20$ | $0.13 \pm 0.09$ | $0.59 \pm 0.13$ | $0.55 \pm 0.18$ | $0.56 \pm 0.17$ | $0.62 \pm 0.12$ |
| | F-MNIST | $0.87 \pm 0.02$ | $0.14 \pm 0.20$ | $0.64 \pm 0.12$ | $0.85 \pm 0.02$ | $0.83 \pm 0.05$ | $0.80 \pm 0.06$ | $0.84 \pm 0.02$ |
| | MNIST | $0.92 \pm 0.03$ | $0.15 \pm 0.20$ | $0.36 \pm 0.14$ | $0.92 \pm 0.03$ | $0.87 \pm 0.08$ | $0.74 \pm 0.12$ | $0.88 \pm 0.03$ |
| **Text** | TREC | $0.41 \pm 0.07$ | $0.15 \pm 0.04$ | $0.27 \pm 0.09$ | $0.40 \pm 0.08$ | $0.37 \pm 0.11$ | $0.23 \pm 0.08$ | $0.41 \pm 0.09$ |
| | AG News | $0.76 \pm 0.08$ | $0.24 \pm 0.02$ | $0.36 \pm 0.10$ | $0.68 \pm 0.08$ | $0.65 \pm 0.08$ | $0.64 \pm 0.10$ | $0.68 \pm 0.10$ |
| | DBpedia | $0.64 \pm 0.19$ | $0.07 \pm 0.02$ | $0.15 \pm 0.08$ | $0.57 \pm 0.19$ | $0.53 \pm 0.19$ | $0.44 \pm 0.21$ | $0.56 \pm 0.17$ |
| | IMDB | $0.62 \pm 0.04$ | $0.50 \pm 0.01$ | $0.50 \pm 0.01$ | $0.59 \pm 0.04$ | $0.58 \pm 0.04$ | $0.57 \pm 0.04$ | $0.60 \pm 0.04$ |

Table 6: Cross-architecture stitching with various methods for estimating $\mathcal{T}$ and applying L2 as normalization. The stitched decoders are simple MLPs. 5 runs for each encoder-decoder pair. (C) and (F) next to `CIFAR100` indicate, respectively, coarse-grained and fine-grained.

| | Dataset | No Stitching | absolute | relative | affine | linear | l-ortho | ortho |
|---|---|---|---|---|---|---|---|---|
| **Vision** | CIFAR10 | $0.95 \pm 0.03$ | $0.16 \pm 0.22$ | $0.73 \pm 0.21$ | $0.93 \pm 0.04$ | $0.89 \pm 0.11$ | $0.89 \pm 0.11$ | $0.93 \pm 0.04$ |
| | CIFAR100-C | $0.82 \pm 0.07$ | $0.11 \pm 0.21$ | $0.39 \pm 0.17$ | $0.77 \pm 0.07$ | $0.75 \pm 0.13$ | $0.71 \pm 0.15$ | $0.78 \pm 0.06$ |
| | CIFAR100-F | $0.68 \pm 0.14$ | $0.06 \pm 0.20$ | $0.13 \pm 0.09$ | $0.60 \pm 0.12$ | $0.57 \pm 0.18$ | $0.54 \pm 0.18$ | $0.61 \pm 0.12$ |
| | F-MNIST | $0.87 \pm 0.02$ | $0.14 \pm 0.20$ | $0.64 \pm 0.12$ | $0.86 \pm 0.02$ | $0.79 \pm 0.09$ | $0.83 \pm 0.05$ | $0.84 \pm 0.02$ |
| | MNIST | $0.92 \pm 0.03$ | $0.15 \pm 0.20$ | $0.36 \pm 0.14$ | $0.91 \pm 0.03$ | $0.80 \pm 0.17$ | $0.86 \pm 0.08$ | $0.86 \pm 0.04$ |
| **Text** | TREC | $0.41 \pm 0.07$ | $0.15 \pm 0.04$ | $0.27 \pm 0.09$ | $0.51 \pm 0.06$ | $0.27 \pm 0.10$ | $0.47 \pm 0.13$ | $0.49 \pm 0.06$ |
| | AG News | $0.76 \pm 0.08$ | $0.24 \pm 0.02$ | $0.36 \pm 0.10$ | $0.68 \pm 0.08$ | $0.64 \pm 0.10$ | $0.65 \pm 0.08$ | $0.66 \pm 0.10$ |
| | DBpedia | $0.64 \pm 0.19$ | $0.07 \pm 0.02$ | $0.15 \pm 0.08$ | $0.55 \pm 0.19$ | $0.53 \pm 0.21$ | $0.51 \pm 0.18$ | $0.49 \pm 0.15$ |
| | IMDB | $0.62 \pm 0.04$ | $0.50 \pm 0.01$ | $0.50 \pm 0.01$ | $0.60 \pm 0.04$ | $0.58 \pm 0.04$ | $0.59 \pm 0.04$ | $0.59 \pm 0.04$ |

## A.1 Scale invariance

In this section, we delve into the concept of scale invariance in neural networks and its implications for model compositionality. We start by focusing on the effect of rescaling operations on the latent input encodings and demonstrate that, by construction, certain classifiers exhibit scale-invariance properties without the need for additional priors. Then, by examining the behavior of networks when subjected to a specific type of input manipulation, *rescaling injection*, we aim to demonstrate the robustness and versatility of neural networks in handling different scales of input data. As illustrated in the main manuscript, this is a key advantage in improving the adaptability of our method.

The softmax function, commonly used in neural classifiers, is known to be a temperature-controlled variant of the maximum function:

$$\text{softmax}(x)_i = \frac{e^{\frac{y_i}{T}}}{\sum_j^N e^{\frac{y_j}{T}}} \,. \tag{2}$$

This means that the softmax temperature can be used to control the level of confidence of the classifier's predictions. In this study, we show that a similar effect can also be achieved by rescaling the latent encodings given as input to a trained (and frozen) classifier.

In order to demonstrate this, we first note that the rescaling factor, $\alpha$, can be factored out of the matrix multiplication in the Linear layers of the classifier. This can be represented mathematically as: $\mathbf{y} = \alpha \mathbf{W} \mathbf{x} + b$, where $\mathbf{x}$ is the input latent encoding, $\mathbf{W}$ is the weight matrix, $b$ is the bias vector, $\alpha$ is the rescaling factor, and $\mathbf{y}$ is the output of the linear layer. This implies that the rescaling operation can be "pushed through" the classifier without affecting its final prediction as it becomes equivalent to some temperature value applied at the softmax level.

Furthermore, we investigate the effect of rescaling when non-linear activation functions are involved and posit that as long as the function has a monotonic interval, if we rescale all the dimensions by an amount similar to the mean scale of the encodings on which the classifier was trained, we end up in the monotonic interval, without losing the scale-invariance property.

In summary, our study provides empirical evidence that neural classifiers that utilize the softmax activation function can, in practice, maintain their scale-invariance properties when the input latent encodings are rescaled. This property is essential to our method, as it allows us to ignore the exact scale when decoding toward an L2-normalized absolute space.

**Pre-trained models and scale-invariance**    We observed that large pre-trained models, such as transformers and resnets, are robust to internal rescaling of the encodings. Although we do not have a strong theoretical explanation for this phenomenon, we hypothesize that normalization layers and the linear separability of the information encoded in the angles instead of the norms may play a significant role. In Figure 12, we demonstrate the invariance a large transformer exhibits when the rescaling injection is applied at different layers: surprisingly, when the rescaling surpasses a certain threshold, the performance difference becomes negligible. These results further emphasize the robustness of these pre-trained models to the rescaling injection and suggest that the scale of the embedding is not a critical factor in their performance.

**Rescale Injection**    We define the *rescaling injection* as the operation of artificially altering the scale of the features produced at a specific layer of the network. This is achieved by normalizing the embeddings to unit norm and then rescaling them by a factor of $\alpha$. By varying the value of $\alpha$, we can observe how the network's performance is affected at different scales. Through this empirical analysis, we aim to provide insight into the scale invariance properties of neural networks and their potential for use in model compositionality.

In Figure 13, we present experimental results investigating the scale invariance properties of neural networks. We trained simple multi-layer perceptrons (MLPs) composed of two hidden layers, with no normalization layers, using encodings produced by the Clip Vision transformer (`clip-vit-base-patch32`) on the `CIFAR100` (fine) dataset. The MLPs were evaluated using different activation functions: cosine (blue), tanh (orange), and ReLU (green). The rescaling injection technique was applied directly to the input embeddings, rescaling them by $\alpha$.

We can observe that the scale of the embeddings does not significantly impact the MLPs' performance when using monotone activation functions that do not flip signs. This is a non-trivial result, as the nonlinearity of the activation function, the presence of bias terms $b$, and the absence of normalization layers make it difficult to predict the effect of an input rescaling on the performance of the network. It is particularly interesting to see that the cosine activation function shows an oscillatory performance, comparable to the original embeddings when rescaled by the mean embeddings scale (vertical red line) or its opposite since it is symmetric.

Our findings indicate that, surprisingly, even the internal layers of large deep learning models exhibit a *positive scale invariance*, as illustrated in Figure 12. The underlying mechanism for this behavior is not straightforward, but we hypothesize that it may result from the interplay between various factors,

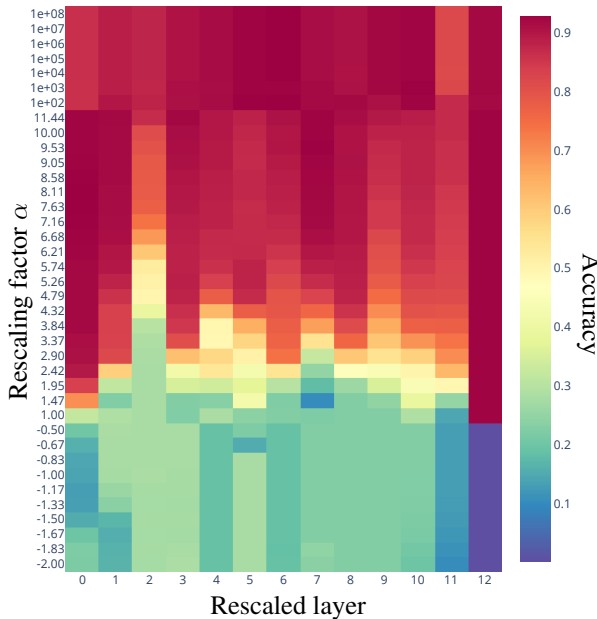

Figure 12: Scale invariance of RoBERTa according to the performance of a downstream classifier trained on the encodings of the last attention layer. At each layer (with 0 being the embedding layer and 12 the output one), one for each run, we rescale the encodings by the specified $\alpha$ and measure its effect on the final accuracy. The performance without any rescaling is $0.92$.

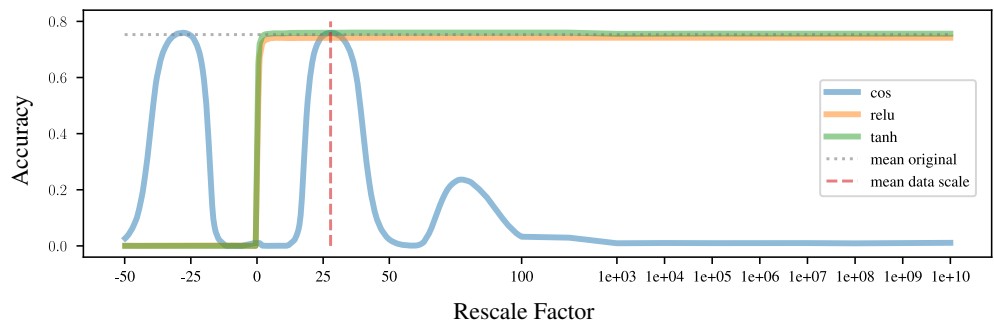

Figure 13: Performance comparison of three Multilayer Perceptrons (MLPs) with different activation functions, namely cosine (blue), ReLU (orange), and tanh (green) at different rescaling factors $\alpha$. The ReLU and tanh MLPs exhibit scale invariance, while the cosine activation function is only invariant on the mean data scale and its periodic cycles.

such as the choice of activation function, the use of normalization layers, the optimization objective and regularization techniques employed during the training phase. Further research is needed to understand and explain this phenomenon fully.

## B   Implementation Details

All the experiments were conducted using a machine equipped with an Intel Core i7-9700k CPU, 64 GB of RAM, and an NVIDIA 2080TI GPU.

**Decoder structure**   The full implementation details can be found in the attached code. The various experiments can be run by their corresponding notebook, while the source code for the package they are built on can be found under the "src" folder.

- *Autoencoding.* Since the autoencoders were used only on image data, the architecture was a simple sequence of convolutions (in the encoder part) and deconvolutions (in the decoder part). Each interleaved with nonlinear activations.
- *Classification.* The main manuscript refers to "SVM" as the standard SVM implementation in scikit-learn [Pedregosa et al., 2011], with default parameters. The experiments with "MLP" as a classifier refer to a simple stack of 3 linear layers, interleaved by nonlinear activations.

**Software and Technologies**   The research of this study was facilitated by the use of various technologies and tools, which include:

- *NN-Template* [GrokAI, 2021], was used to kick-start the project while also ensuring best practices were adhered to;
- *DVC* [Kuprieiev et al., 2022], was implemented for data versioning;
- *PyTorch Lightning* [Falcon and The PyTorch Lightning team, 2019], contributed to maintaining the integrity of the results and promoting clean, modular code;
- *Weights and Biases* [Biewald, 2020], were employed for logging experiments, running comparisons over extensive sweeps, and sharing models;
- *Transformers by HuggingFace* [Wolf et al., 2020], provided pre-configured transformers for processing both image and text data;
- *Datasets by HuggingFace* [Lhoest et al., 2021], facilitated access to a majority of NLP datasets and ImageNet for computer vision purposes;

**Pre-trained encoders**   All the pre-trained encoders used come from HuggingFace and are listed in Table 7. They are various both in terms of architecture and encoding size.

Table 7: HuggingFace models used as encoders (feature extractors) in the various experiments, with their encoding dimensionality.

| Modality | HuggingFace model name | Encoding Dim |
|---|---|---|
| Language | bert-base-cased | 768 |
| | bert-base-uncased | 768 |
| | google/electra-base-discriminator | 768 |
| | roberta-base | 768 |
| | albert-base-v2 | 768 |
| | xlm-roberta-base | 768 |
| | openai/clip-vit-base-patch32 | 768 |
| Vision | rexnet_100 | 1280 |
| | cspdarknet53 | 768 |
| | vit_small_patch16_224 | 384 |
| | vit_base_patch16_224 | 768 |
| | vit_base_patch16_384 | 768 |
| | vit_base_resnet50_384 | 768 |
| | openai/clip-vit-base-patch32 | 768 |

Table 8: Cross-architecture stitching for reconstruction tasks. 5 different seeds, 2 different bottleneck sizes (250, 500) for the same architecture. Average over all combinations. 500 anchors used and standard scaling as normalization. The naive absolute baseline is impossible to compute due to the dimensionality mismatch.

| | MNIST | | | Fashion MNIST | | | CIFAR10 | | | CIFAR100 | | |
|---|---|---|---|---|---|---|---|---|---|---|---|---|
| | *lcos* | *lmse* | *rmse* | *lcos* | *lmse* | *rmse* | *lcos* | *lmse* | *rmse* | *lcos* | *lmse* | *rmse* |
| affine | 0.95 | 0.09 | 0.02 | 0.95 | 0.09 | 0.04 | 0.98 | 0.06 | 0.05 | 0.98 | 0.07 | 0.06 |
| linear | 0.64 | 1.00 | 0.11 | 0.66 | 1.10 | 0.16 | 0.77 | 0.60 | 0.16 | 0.78 | 0.52 | 0.16 |
| l-ortho | 0.87 | 0.16 | 0.03 | 0.89 | 0.14 | 0.06 | 0.95 | 0.12 | 0.08 | 0.95 | 0.13 | 0.08 |
| ortho | 0.91 | 0.14 | 0.03 | 0.92 | 0.13 | 0.06 | 0.96 | 0.12 | 0.09 | 0.96 | 0.12 | 0.09 |

