# OpenReview forum: "Latent Space Translation via Semantic Alignment"
_NeurIPS.cc/2023/Conference — NeurIPS 2023 poster_

### Official Review · Reviewer_zvRG · 2023-06-12

**Soundness:** 3 good
**Presentation:** 4 excellent
**Contribution:** 4 excellent
**Rating:** 8
**Confidence:** 3

**Summary:**

This work proposed a method to translate learned representations between two pre-trained networks, using surprisingly simple transformations. The method is demonstrated on models with different architectures, trained on different modalities and across different tasks.



**Strengths:**

1. The paper is well written and easy to follow.
2. To my understanding, the discussed setting where models are ad-hoc zero-shot stitched, as opposed to being trained to have compatible representations is original. This distinction is important as it makes the approach applicable for the myriad of pre-trained models that already exist.
3. The method is surprisingly simple and yet produces very convincing results as demonstrated through sufficient evaluation.


**Weaknesses:**

1. The exact setting leading to Figure 2 is not clear. Caption mentions results being produced on “CIFAR-100”. However, I was not able to find the entire setting. What two models were used? Are both trained on CIFAR-100? What are the architectures used? Are reported results averaged across multiple settings?
2. Also regarding Figure 2 - the two trends not being monotone is surprising to me. Especially the very clear outliers. I suppose this behavior might occur if each point represents a single experiment and some randomness might affect performance. In that case, it would probably be better to average these results across several experiments as done in other tables in the paper. Can the authors explain this behavior?


Minor comments:

1. Typo line 49 - mphasizing -> emphasizing.
2. Notation line 139 - $\mu$ is used in equation 3 but $\bar{x}$ in text.
3. Clarity line 167 - I suggest the “naive absolute baseline” should be concisely explained.


**Questions:**

See questions mentioned in the Weaknesses section.

**Limitations:**

Yes.

---

> ### Author Rebuttal · Authors · 2023-08-10
>
> We thank the reviewer for appreciating our work, providing valuable feedback, and firmly pushing for its acceptance.
>
> - **Figure 2 (Additional Details)**: In response to the concern about missing details, we will provide comprehensive information in the appendix. For the classification aspect (left), the results are derived from averaging across all potential stitching combinations (pairs) among the utilized pre-trained encoders and their classification heads (SVMs) trained on CIFAR100. With a total of 7 distinct encoders (refer to the appendix, Table 4), this results in a pool of 42 pairs for each "number of anchors" configuration. Regarding the generation aspect (right), we train vanilla convolutional autoencoder pairs on the CIFAR100 dataset for each "number of anchors" configuration. No normalization is applied to the dataset images, and each autoencoder has ~2 million trainable parameters and a size 500 bottleneck.
> - **Figure 2 (Trends)**: The presence of outliers and the non-monotonic pattern observed in the lstsq curve can be attributed to a confluence of factors, including numerical instability and the PyTorch **driver selection** policy. This policy adapts based on the condition number of the matrix. In particular, we observed that outliers tend to emerge as the amount of anchors approaches the number of dimensions, leading to a square matrix.
> - **Naive Absolute Baseline**: this baseline entails the direct stitching between models considering their vanilla representations without any transformation, to verify any pre-existing compatibility and establish it as a lower-bound. We agree it deserves further explanation, we will integrate this description in the preliminaries, since it is also used in Moschella et al..
>
> **Figure 2 Updates**
>
> In response to these observations, we systematically repeated all classification experiments using five distinct random seeds, influencing initialization and anchor selection. Additionally, we have transformed the plot into a violin chart format to present the variance clearly. We will do the same for the generation experiments. The updated visualization we will include in the paper can be found in the rebuttal PDF.

---

> > ### Comment · Reviewer_zvRG · 2023-08-13
> >
> > Thanks for the rebuttal! It addresses the comments I've raised in my review.

---

### Official Review · Reviewer_rL3F · 2023-07-05

**Soundness:** 3 good
**Presentation:** 3 good
**Contribution:** 3 good
**Rating:** 6
**Confidence:** 4

**Summary:**

In this paper the authors propose to translate latent space using an effectively angle-preserving affine transformation learned using pairs as anchors. Experiments are conducted on a wide range of tasks showing the okay performance by the swap-in latent space of embeddings/features.

**Strengths:**

**Strengths**

The authors have taken an extensive suite of experiments encompassing a diverse set of tasks, including cross-training, cross-architectures, and several downstream tasks. The experimental design demonstrates a comprehensive approach in assessing the proposed method *per se*.

Furthremore, this work shows that aligning two latent spaces with zero-shot stitching is poissible without any retraining. This is an important observation and the proposed method could benefit a wide range of downstream tasks.

**Weaknesses:**

**Weaknesses**

The following concerns regarding the design of the transformation and its intrinsic limitation warrant consideration.

- **Assumption of Linearity and Lack of Analysis in Generative Model**:


The proposed method is effectively an angle-preserving affine transformation, which fundamentally assumes linearity of mapping between two latent spaces (of embeddings and/or features). While "linearity of the transformation between seemingly different latent spaces has a theoretical foundation in research on identifiability in neural models," as the author pointed out in the rebuttal, in the context of generative model, the efficacy is still likely contingent on the extent to which two latent spaces exhibit linear mapping, because a good generative model may fill the slight mismatch with its own capacity.

In detail, it is likely that an affine transformation may yield only a coarse mapping between two latent spaces, and may not capture fine-grained relations (consider style transfer in generative models).
Thus, the good performance in classification tasks may thus be attributed to the capacities of the downstream models rather than the transformation itself. This is a critical assumption that necessitates further scrutiny, but the submission does not provide an in-depth analysis of this assumption.

In the current form of the manuscript, for this regard only an autoencoding on MNIST/Fashion MINIST/CIFAR is shown. It does not have a qualitative comparison with the reconstruction from VAE itself, nor quantitative/qualitative results with other stitching and zero-shot methods.
The lack of such comparison can unfortunately weaken the justification of the proposed method.
Adding such comparison is thus suggested. Optionally, conducting such comparisons on common image datasets of higher resolution, like CelebA, ImageNet, (with more powerful VAE models) may provide more insight in this regard, since it may potentially highlight the capacity of VAE models and the quality of latent space alignment.

(**After author response**: See the discussion below)

- **Ambiguity in Technical Contribution**:

The technical novelty and significance of the proposed method remain unclear. The method effectively is an angle-preserving affine transformation. While this per se is not necessarily an issue, the missing studies mentioned above make it hard to assess the impact and implications of the proposed method.

(**After author response**: This is not a major issue anymore, see the discussion below)

- ** Position of Works within the Broad Field**:

Latent space matching has been the subject of extensive research, albeit in varying contexts such as text, image processing, and generative models.
A few (it's even not an attempt to cover the overall picture of this widely studied topic) works below in References that share technical similarity with the proposed method  are cited in the References.
However, the submission does not include a substantive comparison or discussion with a wide body of works in this field, mkaing it challenging to ascertain the relative standing and uniqueness of the proposed method.

(**After author response**: This is not a major issue anymore, see the discussion below)

**References**:

-
Set Prediction in the Latent Space https://proceedings.neurips.cc/paper/2021/hash/d61e9e58ae1058322bc169943b39f1d8-Abstract.html
- Formality Style Transfer with Shared Latent Space https://aclanthology.org/2020.coling-main.203.pdf
- Homomorphic Latent Space Interpolation for Unpaired Image-To-Image Translation https://openaccess.thecvf.com/content_CVPR_2019/html/Chen_Homomorphic_Latent_Space_Interpolation_for_Unpaired_Image-To-Image_Translation_CVPR_2019_paper.html



**Updates**

- The previous version of this review mistakenly refer to the page limit. As the authors and the AC pointed out, this is NOT the case and has been corrected.

- In Weakness, update the Assumption of Linearity to focus on the generative model.

- Added reference to discussion below.

- The rating has been increased.

**Questions:**

N/A

**Limitations:**

Yes

---

> ### Author Rebuttal · Authors · 2023-08-10
>
> We thank the reviewer for the review and comments. We appreciate the opportunity to address the concerns they've raised and offer clarifications on the following points:
>
> **Page Limit**: As per the **official 2023 Call for Paper**, submissions are indeed limited to **nine** content pages and **not eight**. Therefore, we don’t see any reason to report the paper to be desk-rejected.
>
> **Assumption of Linearity and Lack of Analysis**:
>
> - The assumption of linearity of the transformation between seemingly different latent spaces has a theoretical foundation in research on identifiability in neural models [a, b, c,e,f]. In particular, in [a], it is proved that representations learned by a large family of supervised and generative models are identifiable up to a linear transformation. We will incorporate these additional references in the related works section. Our work is intrinsically centered around exploring the emergence of these properties in practical settings within deep neural networks (DNNs).
> - While we acknowledge the broader context of latent space matching, the references the Reviewer kindly suggested involve the introduction of constraints, penalties, or adjustments *during training*, which lie outside the scope of our investigation.
> - Our focus, instead, aligns closely with the works highlighted in the "Stitching and Zero-Shot" section of our manuscript (Section 2). These works exploit the linearity assumption and focus on finding a transformation that aligns the latent spaces ex-post. For example, in a very recent work, Moayeri et al. [d] further prove that these emerging properties exist, can be exploited, and deserve more attention from the community.
>
> [a] Roeder, Geoffrey, Luke Metz, and Durk Kingma. "On linear identifiability of learned representations." *ICML*, PMLR, 2021.
>
> [b] Khemakhem, Ilyes, et al. "Ice-beem: Identifiable conditional energy-based deep models based on nonlinear ica.", NeurIPS, 2020
>
> [c] Willetts, Matthew, and Brooks Paige. "I Don't Need u: Identifiable Non-Linear ICA Without Side Information." ArXiv
>
> [d] Moayeri et al. “*Text-To-Concept (and Back) via Cross-Model Alignment*”*,* ICML 2023
>
> [e] Hyvarinen et al "Nonlinear ICA using auxiliary variables and generalized contrastive learning."  PMLR, 2019.
>
> [f] Sorrenson,et al. “Disentanglement by nonlinear ica with general incompressible-flow networks (gin)” 2020
>
> **Coarse-only mapping:** We would like to emphasize that we worked with both classification and generation as decoder tasks, observing good performance even in the latter scenario. This contrasts with the following comment: “*an affine transformation may yield only a coarse mapping between two latent spaces, and may not capture fine-grained relations*”. We will be happy to include other datasets/tasks to broaden our study, following any suggestions from the Reviewer.

---

> > ### Comment · Reviewer_rL3F · 2023-08-15
> >
> > I would like to first thank the authors and the AC for pointing out the page-limit. I Also appriciate the authors' clarification on linearity and zero-shot stitching, and agree with that. Both have been reflected in the revised review.
> >
> > My concern on assumption of linearity and lack of analysis is on generative models. While it's clear, as the authors pointed out, that for calcification tasks linearity of representation has been identified with theoretical foundation, I'm still not fully convinced of the degree of such a case for generative models. Please see the revised review for details on this regard.

---

> > > ### Author Response · Authors · 2023-08-18
> > >
> > > We thank the Reviewer for taking the time to reconsider their evaluation and adjust their review, particularly concerning page limit, linearity, and zero-shot stitching.
> > >
> > > We respectfully reiterate our answer to the *concern of Insufficient Comparison with Works within the Broad Field*. We will be happy to discuss the references provided by the Reviewer in the related work section. However, we still consider them out-of-scope for an experimental comparison: **the referenced methods require the enforcement of additional constraints during training.** In contrast, **our post-hoc method can be applied directly to pre-trained models**.
> > >
> > > Regarding the *assumption of linearity in generative models*:
> > >
> > > There is a large collection of results in the literature on the identifiability of generative models with auxiliary information [b, c, d, e, f, g, h, i]. When this side information is not available, it is not possible to have strict theoretical guarantees, as shown in [a] and [m]. However, a recent line of work demonstrated that it is possible to have the same characterization in unsupervised generative models, either by providing experimental evidence  [Moschella et al 2022, p], via measuring high identifiability scores for unsupervised generative models [n] or by providing theoretical evidence with a weaker notion of identifiability [p,o], but no auxiliary information. Our experimental assumptions are supported by these findings, suggesting that **in most cases** it is possible to connect latent spaces of generative models via simple transformations.
> > >
> > > We believe that our method's simplicity is a strength, not a weakness, of our work. The novel insights and practical applicability of our findings contribute to advancing the field, and we are committed to sharing this knowledge with the community.
> > >
> > > Once again, we thank the Reviewer for their valuable feedback and remain available to address any additional queries.
> > >
> > > **[a]** Francesco Locatello, Stefan Bauer, Mario Lucic, Gunnar Rätsch, Sylvain Gelly, Bernhard Schölkopf, Olivier Bachem. Challenging Common Assumptions in the Unsupervised Learning of Disentangled Representations. ICML 2019
> > >
> > > **[b]** A. Hyvärinen, H. Sasaki, and R. E. Turner. Nonlinear ICA Using Auxiliary Variables and Generalized. PMLR 2019
> > >
> > > [**c**] P. Sorrenson, C. Rother, and U. Köthe. Disentanglement by Nonlinear ICA with General Incompressible-flow Networks (Gin). ICLR 2020
> > >
> > > [**d**] Khemakhem, R. P. Monti, D. P. Kingma, and A. Hyvärinen. ICE-BeeM: Identifiable Conditional Energy-Based Deep Models Based on Nonlinear ICA. NeurIPS 2020
> > >
> > > [**e**] A. Hyvärinen and H. Morioka. Unsupervised Feature Extraction by Time-Contrastive Learning and Nonlinear ICA. NeurIPS 2016
> > >
> > > [**f**] Locatello, Francesco, et al. "Weakly-supervised disentanglement without compromises." International Conference on Machine Learning. PMLR, 2020.
> > >
> > > [**g**] Locatello, Francesco, et al. "Disentangling Factors of Variation Using Few Labels”. ICLR 2020
> > >
> > > [**h**] Khemakhem, Ilyes, et al. "Variational autoencoders and nonlinear ICA: A unifying framework." International Conference on Artificial Intelligence and Statistics. PMLR, 2020.
> > >
> > > [**i**] Von Kügelgen, Julius, et al. "Self-supervised learning with data augmentations provably isolates content from style.". NeurIPS 2021
> > >
> > > [**l**] Zimmermann, Roland S., et al. "Contrastive learning inverts the data generating process." International Conference on Machine Learning. PMLR, 2021.
> > >
> > > [**m]** Hyvärinen, et al. "Nonlinear independent component analysis: Existence and uniqueness results." *Neural networks* 12.3 (1999):
> > >
> > > [**n]** Willetts, Matthew, and Brooks Paige. "I Don't Need u: Identifiable Non-Linear ICA Without Side Information." ArXiv
> > >
> > > [**o**] Barin-Pacela, Vitória, et al. "Identifiability of Discretized Latent Coordinate Systems via Density Landmarks Detection.". ICML Workshop on Structured Probabilistic Inference & Generative Modeling, 2023.
> > >
> > > [**p]** Asperti, et al. "Comparing the latent space of generative models." *Neural Computing and Applications* 35.4 (2023)
> > >
> > > [**q]** Kivva, Bohdan, et al. "Identifiability of deep generative models without auxiliary information." NeurIPS 2022

---

### Official Review · Reviewer_NjdJ · 2023-07-06

**Soundness:** 3 good
**Presentation:** 2 fair
**Contribution:** 2 fair
**Rating:** 5
**Confidence:** 3

**Summary:**

This article proposes a method of latent space translation through semantic alignment, using the similarity of latent spaces learned by different neural models on semantically similar data. The method allows for direct conversion of learned representations between different pre-trained networks, and achieve zero-shot stitching of encoders and decoders without additional training. The method is widely validated in various experimental settings, tasks (classification and generation), and modalities, demonstrating the ability to zero-shot stitch neural models under different architecture and modality changes.

**Strengths:**

1. The paper proposes a method for the zero-shot stitching between encoders and decoders on various tasks, enabling seamless integration without the need for additional training.
2. Compared to previous methods, this article expands to different latent space dimensions, architectures, and modalities.
3. Extensive set of experiments have been done to show the possibility to zero-shot together neural models across different architectural and modalities.

**Weaknesses:**

1. In the experiment, only the most basic abs method is compared, but no comparison is made with other methods.
2. In the generation task, zero-shot stitching is performed under the same model framework and training data. Whether it is more practical to migrate between different model frameworks and training data?

**Questions:**

1. How much performance difference is there between the proposed zero-shot stitch method and concatenation followed by finetuning?
2. How is the selection of anchor points determined, and how much impact does different selection have on performance?
3. In experiments, zero-shot stitch is performed between different frameworks on the same dataset. So, can this method be effectively used for neural frameworks trained on different domain datasets?

**Limitations:**

Yes

---

> ### Author Rebuttal · Authors · 2023-08-10
>
> We thank the reviewer for their valuable input, which has prompted us to further enhance the soundness of our work.  Here, we provide deeper clarification and offer additional insights regarding:
>
> **Additional Comparisons**: we will add several columns to our tables (Table 1 and Table 2): **i)** performance of relative decoders; **ii)** stitching performance of relative decoders (to be used as a direct comparison with the Moschella et al. method); **iii)** stitching performance when optimizing an affine transformation with SGD (using the same anchor set as the other methods), namely “Linear”, as a standard and robust alternative stitching method [a, b]. We remain open to additional baseline suggestions.
>
> [a] Lenc and Vedaldi “*Understanding image representations by measuring their equivariance and equivalence.*“CVPR 2015.
>
> [b] Moayeri et al. “*Text-To-Concept (and Back) via Cross-Model Alignment*”*,* ICML 2023
>
> A preview of the new Table 1 can be found in the rebuttal PDF.
>
> **Cross-Architecture stitching (generation)**: in the rebuttal PDF, we show a stitching experiment between the same autoencoder architecture used throughout the whole paper but with varying training seed (2 different ones) and bottleneck size (250 and 500) on multiple datasets. The performances are similar to the ones in Table 3 of the main manuscript, showing that cross-architecture stitching is possible even between autoencoders and has good performances.
>
> **Fine-tuning (Question 1)**: it is unclear to us the link between fine-tuning on a concatenation of the two representations and stitching. What would be the downstream training objective? Could the reviewer kindly expand on this? We’d be open to adding it as an alternative/baseline method.
>
> **Anchor selection (Question 2)**: Our approach adheres to the standard RelRep setting, where anchor points are uniformly drawn from the training samples. Notably, Moschella et al. indicate that more intricate policies do not yield discernible improvement, a finding supported by their supplementary analysis in the appendix (Moschella et al., Section A2). We will further clarify this in the Preliminaries section.
>
> **Multi-domain (Question 3)**: if we interpreted the question correctly, the reviewer is asking whether our method can be applied to translate across samples represented in different domains. We see our “Cross-modality” section (4.2), where we translate from the text domain to the image one and vice-versa, as a generalization of this idea, given that a modality change is supposedly stronger than a domain one. Nevertheless, we present in the rebuttal PDF and will incorporate in the paper a new experiment where we stitch between classifiers trained on different versions of the CIFAR10 dataset: the standard one and a grayscale version. The performances are comparable to the ones in Table 1 of the main manuscript (CIFAR10 row), showing that cross-domain stitching is possible and with good performances.
>
> We thank the reviewer again for their feedback and remain available to discuss any further concerns or questions.

---

> > ### Comment · Reviewer_NjdJ · 2023-08-21
> >
> > Thanks for your rebuttal.

---

### Official Review · Reviewer_ivQN · 2023-07-06

**Soundness:** 3 good
**Presentation:** 1 poor
**Contribution:** 3 good
**Rating:** 6
**Confidence:** 3

**Summary:**

This paper show the latent space of different pretrained models can be translated between each other with simple transformations by using anchors. By using this method, a variety of encoders architectures are able to be cross-stiched to different classifier heads or within autoencoding models.

**Strengths:**

This paper tackles an interesting problem for how to semantically align different trained latent spaces.

The paper is a natural extension to Moschella et. al 2023, going from just relative encoding to full latent space transformations.

The results are clear in showing zero-shot stitching is possible without retraining the decoders

**Weaknesses:**

It wasn't clear on my first read of the paper that this work is a direct follow up to Moschella et al. 2023. Without that context, the paper was very difficult to read the first time. After reading Moschella et al., I was better able to follow this proposed work. Multiple details are missing, such as the construction of the anchor set, which are glossed over in this work and caused confusion on the zero-shot nature of this work. Another major missing key detail is that prior work's decoders must work on the relative embedding, this crucial detail was not clear until after multiple read throughs. This improvement should be front and center.

I find it surprising that tables 1 and 2 do not report Moschella's results. I realize that the decoders must be trained with the relative embeddings, but the lack of comparison feel likes an obfuscation rather than a highlight of the difference between methods.

Overall, I find the clarity of the paper to be a major hinderance.

**Questions:**

None

**Limitations:**

Yes

---

> ### Author Rebuttal · Authors · 2023-08-10
>
> We sincerely appreciate the reviewer's insightful feedback, highlighting important aspects warranting clarification and enhancement in our work. In light of their review, we have identified these steps to improve our work:
>
> - Emphasize the relationship with RelRep in the Preliminaries section as already stated in the **general response** to better contextualize the foundations of our work;
> - Add a direct comparison with the RelRep method in both Table 1 and 2. A preview of the new results to be added to Table 1 can be found in the rebuttal PDF.
> - The anchor set construction procedure, aligned with the standard RelRep sampling (a uniform random sampling over the training set), will be incorporated into the Preliminaries section. This will both elucidate the process and draw a parallel with the original work's findings in Moschella et al..
> - We will offer a more transparent depiction of our work's "zero-shot" nature. As correctly highlighted by the reviewer, our distinction from RelRep lies in removing the requirement for relative training of the decoder. Instead, we achieve (better) stitching performance by solving a closed-form problem. This methodology will be emphasized to highlight our innovation.
>
> We hope to have clarified the reviewer's concerns, and we respectfully remark that R2, R3, and R4 didn't consider clarity a significant weakness, assigning higher scores to it. Nevertheless, we remain available to discuss any further concerns or questions.

---

> > ### Comment · Reviewer_ivQN · 2023-08-21
> > **Thanks authors.**
> >
> > The extra context is great. The clarity issues seem to be due to my unfamiliarity, I will increase my score to weak accept. I still am a little concerned with the final framing, but that's not a content issue.

---

### Author Rebuttal · Authors · 2023-08-10

We sincerely appreciate the constructive feedback provided by the reviewers. We would like to address the relationship between our work and the concepts presented in "relative representations" (Moschella et al.). While we draw upon the foundational principles of Moschella et al., it is essential to clarify that our work takes a different path. Therefore, we do not consider it a direct extension.

- Moschella et al. showed that when different latent spaces share semantics: 1) a simple angle-preserving transformation connects them; 2) a reduced set of key points (anchors) can be used to reconcile them into a **new, distinct (relative) representation**.
- Instead, we directly estimate the **transformation** from one space to another using provided anchor points without relying on an auxiliary representation. A key advantage of this approach is that it circumvents the necessity of training the decoder on relative representations, simplifying the integration of diverse models arbitrarily trained.

Considering the reviewers' valuable input on clarifying this aspect to strengthen its message, we commit to adding a new section to specify the connection to Moschella et al. better. We'll present this relationship accurately and concisely, expanding the “Preliminaries” section, utilizing available space within the nine-page limit.

---

### Decision · Program_Chairs · 2023-09-21

**Decision:**

Accept (poster)

**Comment:**

All reviewers agree that the paper extends existing work nicely, enabling post hoc stitching of trained models. Initial concerns were mainly around comparisons to existing works, presentation clarity and linearity assumptions on the feature space. Authors have addressed the first 2 concerns in the rebuttal. The 3rd concern seems to be not specific to this work, but more broadly to these line of works, which needs to be addressed in future works. Overall reviewers are happy with author's response and I am happy to suggest acceptance. I encourage authors to improve the presentation of the final version as per reviewers suggestions.